# Parallelized Manipulation of Adherent Living Cells by Magnetic Nanoparticles-Mediated Forces

**DOI:** 10.3390/ijms21186560

**Published:** 2020-09-08

**Authors:** Maud Bongaerts, Koceila Aizel, Emilie Secret, Audric Jan, Tasmin Nahar, Fabian Raudzus, Sebastian Neumann, Neil Telling, Rolf Heumann, Jean-Michel Siaugue, Christine Ménager, Jérôme Fresnais, Catherine Villard, Alicia El Haj, Jacob Piehler, Monte A. Gates, Mathieu Coppey

**Affiliations:** 1Laboratoire Physico Chimie Curie, Institut Curie, PSL Research University, Sorbonne Université, CNRS, 75005 Paris, France; maud.bongaerts@curie.fr (M.B.); koceila.Aizel@curie.fr (K.A.); 2Physico-chimie des Électrolytes et Nanosystèmes Interfaciaux, PHENIX, Sorbonne Université, CNRS, F-75005 Paris, France; emilie.secret@upmc.fr (E.S.); jean-michel.siaugue@upmc.fr (J.-M.S.); christine.menager@sorbonne-universite.fr (C.M.); jerome.fresnais@sorbonne-universite.fr (J.F.); 3Laboratoire Physico Chimie Curie, Institut Pierre Gilles de Gène, Institut Curie, PSL Research University, Sorbonne Université, CNRS, 75005 Paris, France; audric.jan@curie.fr (A.J.); Catherine.Villard@curie.fr (C.V.); 4Guy Hilton Research Centre, School of Pharmacy and Bioengineering, Keele University, Stoke-on-Trent, Staffordshire ST4 7QB, UK; t.nahar@keele.ac.uk (T.N.); n.d.telling@keele.ac.uk (N.T.); 5Department of Biochemistry II – Molecular Neurobiochemistry, Faculty of Chemistry and Biochemistry, Ruhr-Universität Bochum, 44801 Bochum, Germany; fabian.raudzus@ruhr-uni-bochum.de (F.R.); sebastian.neumann@ruhr-uni-bochum.de (S.N.); rolf.heumann@ruhr-uni-bochum.de (R.H.); 6Department of Clinical Application, Center for iPS Cell Research and Application (CiRA), Kyoto University, Kyoto 606-8507, Japan; 7Healthcare Technology Institute, Institute of Translational Medicine, University of Birmingham, Birmingham B15 2TT, UK; a.j.el.haj@keele.ac.uk; 8Department of Biology/Chemistry, University of Osnabrück, Barbarastr. 11, 49076 Osnabrück, Germany; Jacob.Piehler@biologie.uni-osnabrueck.de; 9Institute of Pharmacy and Bioengineering, School of Medicine, Keele University, Keele ST5 5BG, UK; m.a.gates@keele.ac.uk

**Keywords:** nanoparticles, magnetic, cell migration, axonal outgrowth, endosomes, cell polarity, live cell imaging

## Abstract

The remote actuation of cellular processes such as migration or neuronal outgrowth is a challenge for future therapeutic applications in regenerative medicine. Among the different methods that have been proposed, the use of magnetic nanoparticles appears to be promising, since magnetic fields can act at a distance without interactions with the surrounding biological system. To control biological processes at a subcellular spatial resolution, magnetic nanoparticles can be used either to induce biochemical reactions locally or to apply forces on different elements of the cell. Here, we show that cell migration and neurite outgrowth can be directed by the forces produced by a switchable parallelized array of micro-magnetic pillars, following the passive uptake of nanoparticles. Using live cell imaging, we first demonstrate that adherent cell migration can be biased toward magnetic pillars and that cells can be reversibly trapped onto these pillars. Second, using differentiated neuronal cells we were able to induce events of neurite outgrowth in the direction of the pillars without impending cell viability. Our results show that the range of forces applied needs to be adapted precisely to the cellular process under consideration. We propose that cellular actuation is the result of the force on the plasma membrane caused by magnetically filled endo-compartments, which exert a pulling force on the cell periphery.

## 1. Introduction

The remote manipulation of cellular functions through smart nanomaterials constitutes an important challenge in fundamental research, as well as in regenerative medicine. For instance, the guidance of cell migration or nerve growth would be instrumental for neuro-therapeutic applications. Indeed, cell-based therapy stands out as the most promising approach to regenerate disrupted neuronal circuits, but one of the key challenges associated with it is controlling the behavior of fresh neuronal cells transplanted in the body. As nerve regeneration in the central nervous system is limited in the adult body due to the growth inhibitory environment and loss of guidance cues, there is a need for methods to direct axonal regrowth of transplanted neurons toward the disconnected target region, and thus to restore proper neuronal circuitry [1].

Optogenetics tools, based on light-induced activation of biochemical reactions, have been proposed as a promising method to remotely control cell migration [2,3,4,5], as well as to activate fiber outgrowth [6,7]. However, therapeutic applications of optogenetic technology are limited by the poor penetration of light into the body and by the potential cytotoxicity of optical stimuli [8]. In contrast, tools based on the use of magnetic fields offer the advantage of propagating through tissues and organisms without interfering with biological samples, except for very specific settings such as high frequency fields that can heat the sample. Smart magnetic nanomaterials have been designed to actively respond to external magnetic fields, and to act as nano-actuators, able to mediate or convert different forms of energy into both physical and chemical cues, guiding specific cell behavior. Such nanomaterials can manipulate cellular events at distance, and at spatial and temporal scales matching their physiological dynamic range. Moreover, magnetic fields provide multiple modes to control cellular functions, via mechanical actuation, heating or by modulating protein concentrations within cells, making magnetic manipulation particularly promising for therapeutic applications [9].

One of the state-of-the-art techniques to control cell behavior is called “magneto protein therapy” [1], also sometimes referred to as magnetogenetics, the principle of which is to actuate biological processes directly inside the cell by hijacking endogenous signaling pathways. In this context, to be efficient and specific remote-activators within the cell, magnetic nanoparticles (MNPs) require both a remarkable passivation to avoid non-specific interactions and adopt free diffuse behavior, and a remarkable functionalization to target specific proteins inside the complex cell interior [10]. Such engineered MNPs have been used to modulate the spatial distribution of signaling molecules involved in cellular polarity, thus enabling the local formation of membrane protrusions [11]. Recent studies have proposed promising strategies with MNPs designed to activate specifically axonal growth in reprogrammed neurons from induced pluripotent stems cells [12,13]. However, these innovative approaches are very complex and will need further investigations before being really effective for therapeutic applications.

On the other hand, the non-specific mechanical approach constitutes a much more advanced magnetic mode for remotely guiding cell motility and nerve growth, albeit not as specific as the engineered MNPs. As a matter of fact, MNPs have long been used to exert tensile forces on cells, tissue or substrates, and it is well established that physical forces play a key role in cell polarity, migration and in shaping neuronal structure [14]. During migration, the coordinated assembly of multiple actin filaments produces protrusive forces that drive the extension of the plasma membrane (protrusions) at the cell leading edge [15]. Thus, membrane localized forces, applied asymmetrically within the cells, appear to be crucial to determine cell polarity. Mechanical tension is also a particularly effective stimulus for axonal elongation, the process by which neurons connect with each other and form the complex neuronal networks throughout the body. During neuronal development, axons grow in two different phases, both of which are distinguished by the nature of the forces that drive growth. In the first phase, growth cones, the mobile tips of axons, actively exert pulling forces on the processes [16]. In a second phase, after connecting with their target tissue, axons may be passively pulled by the increasing distance between the target and nervous tissue, resulting in considerable growth in length, a process referred to as stretch growth [17].

Over the last decades, many works have revealed the effect of mechanical stimuli on cellular polarity, but the mechanism by which forces are integrated into biochemical processes remains largely unknown. These mechanical signals can come from the extracellular environment, as well as from inside the cells, but can also emerge as the interplay between environmental mechanical properties and forces exerted by cells. For instance, some studies demonstrated that migration of cultured cells can be directionally controlled by modulating substrate stiffness and strain [18,19] or that axonal growth can be induced by artificially stretching a neurite using a glass tip [20]. One promising approach in the field of tissue engineering is the development of scaffolds containing aligned fibers along which cells or nerves can be guided [21,22,23]. Here the cells are stimulated by external physical cues coming from topographical features. Interestingly, a group has designed an injectable and magnetoceptive biomaterial which can be structured remotely with guide fibers by MNPs contained in the gel [24,25]. Taking advantage of these forces by controlling them precisely through smart nanomaterials should therefore bring subsequent benefits in regenerative medicine and in understanding biomechanical aspects.

MNPs have been employed to apply physiological forces to cells as they can be easily remotely controlled in time and space under a magnetic field gradient. MNPs, as force-mediating objects, can manipulate cell structures either inside the cytosol or outside of the cell via the cell membrane. Previous studies have used them with success to manipulate the directional motility of cells [26,27,28,29,30] or neurite elongation, either associated with the cell outer membrane [31] inside the cells, packed into endosomes or as free particles in the cytosol [32,33,34,35].

Tuning the forces precisely is important, as these forces can promote or block healthy cell function depending on their magnitude, direction and duration of application. Kunze et al. characterized the force needed to induce cellular responses by the use of magnetic chip and cells containing magnetic nanoparticles contained within endosomes. Their work revealed that the amplitude of forces required to trigger migration is around hundreds of piconewtons (pN), whereas neurite initiation requires lower forces, in the range of tens of pN. In this study, Tau protein repositioning, which is a specific marker of axonal outgrowths, proved that early axonal elongation can be triggered by mechanical tension inside cells. Raffa et al. confirmed that pN forces applied along the whole axon and over a long period of time can promote statistically oriented outgrowth. These promising results demonstrate that intracellular mechanical forces can be successfully used to drive cell polarity.

However, these prior convincing studies lack real-time observations of live cells, in which the movement of MNPs inside cells can be correlated with the cell displacement or neurite outgrowth over several hours or days. To fill this gap, we utilized a recently developed parallelized magnetic system [36] in order to simultaneously manipulate and image hundreds of living cells over extended periods of time (0–48 h). We were able to remotely control force-induced processes such as cellular migration and neurite outgrowth, by applying mechanical tensions on endosomes filled with MNPs. This system, which combines a micro-magnetic array (MMA) and biocompatible superparamagnetic iron oxide nanoparticles (SPION) internalized in cells, was tested on different cellular models. Our results reveal that membrane-localized mechanical stimuli clearly affect directional motility of adherent cells in a reversible manner, and that the manipulation of MNP-filled endosomes does not disrupt vital cellular functions. Indeed, for the first time, we demonstrate the trapping and subsequent release of adherent cells on magnetic pillars. Most significantly, we demonstrate that the remote magnetic manipulation and accumulation of endosomes at the plasma membrane of a neuronal cellular body can be used to direct neurite initiation and growth, possibly of axons, while preserving cell integrity.

## 2. Results 

### 2.1. A Parallelized Magnetic Tool to Actuate Migration and Neurite Outgrowth through Forces

Magnetic control experiments were performed on four types of cells—human cervix adenocarcinoma HeLa cell lines and human neuroblastoma SHSy-5Y cell lines were used as two distinct “migration” models based on their size, shape and motility behavior. Nerve growth factor responsive rat pheochromocytoma PC12 cells and mouse primary cortical neurons were selected as models for the observation of neurite outgrowth, possibly being axons or dendrites. Fluorescent superparamagnetic Fe_2_O_3_-PAA_2k_-rhodamine nanoparticles (Appendix A) were chosen, as they demonstrated remarkable stability and dispersity in various types of media, biocompatibility and a good internalization efficiency into cells. The PAA coating acts as an efficient protective layer that provides long-lasting cell magnetization [37]. MNPs are effectively internalized by lymphoblastic cells and are visible in endosomes after 24 h of incubation at a concentration of 10 mmol/L [38]. This internalization protocol was used for HeLa cells with success but had to be slightly adapted for the more sensitive and smaller SHSy-5Y, PC12 cells and cortical neurons. Excellent internalization efficiency was observed in the four cellular types, reaching 100% MNP-labelling of cells but with a relatively variable loading of MNPs per cell (Appendix A). Interestingly, we observed that in MNP-loaded PC12 cells, treatment with nerve growth factor (NGF) for 2–9 days resulted in an accumulation of MNPs in 92% of the neurite tips, with varying length, even without magnetic manipulation (Appendix A). The hydrodynamic diameter of Fe_2_O_3_-PAA_2k_-rhodamine MNPs is about 27 nm, such that many of these MNPs can accumulate within endosomes which are ~400 to 600 nm in size, as observed in TEM images performed on MNP-loaded SHSy-5y cells (Appendix A). Thus, MNP-filled endosomes can be considered equivalent to large superparamagnetic nanoparticles, with a corresponding scaling of magnetic forces compared to the individual MNPs.

The micro-magnetic array (MMA) was designed by our group (Figure 1a–d), specifically for the manipulation of small MNPs (magnetic core of <10 nm) inside living cells, by engineering micro-magnets able to promote high magnetic field gradients (10^3^ to 10^4^ T/m) on ~50 µm spatial length scales [36]. Following characterization of both the micro-magnetic pillars (Appendix A) and Fe_2_O_3_-PAA_2k_-rhodamine MNPs (Appendix A), the force acting on one single particle was calculated to be 1 fN at a 5-µm distance from the edge of the micro-pillar (reducing to 0.35 fN at a distance of 10 µm). Based on TEM images of MNP-loaded SHSy-5y cells (Appendix A), we estimated that the corresponding force acting per cell was 22 ± 16 pN at a 10-µm distance from the micro-pillar (summing all nanoparticles inside endosomes and summing all endosomes in a cell; see calculation in Materials and Methods and table in Appendix A).

### 2.2. MNPs Are Stable in Lysosomal Compartment up to Several Days

Regardless of whether MNPs take the clathrin-dependent pathway, clathrin-independent pinocytosis or macropinocytosis, the delivery of the MNPs from the initial compartment to the lysosomes is a rapid process which generally takes less than 30 min. Considering this timescale, by the time the magnetic control experiment started (3 h after removing excess particles), the particles that had associated with cells were expected to be captured into lysosomes. The kinetics of degradation of molecules in lysosomes is very variable and depends on molecule composition. In the case of magnetic nanoparticles, the protective layer which coats the magnetic core must provide a compromise between stability and biodegradability. However, the rhodamine molecules grafted onto this surface could be more exposed to lysosome hydrolases.

To study the fate of Fe_2_O_3_-PAA_2k_-rhodamine nanoparticles in the cell, two experiments were carried out. In first of these, the MNPs’ fluorescence signal after endocytosis was evaluated over 20 days in SHSY-5y cells (Appendix A). Surprisingly, the rhodamine fluorescent signal remained stable for at least 10 days without showing a significant decrease. At 14 days, the signal began to decrease and disappeared almost completely after 20 days. However, it is not surprising to observe a decrease in the fluorescent signal as the number of particles per cell is diluted over cell divisions. Thanks to their PAA coating, the magnetic nanoparticles should be protected from degradation for at least for the first 10 days (Appendix A). Since the metallic core of MNPs is less exposed, their magnetic stability and efficiency should also be preserved.

In the second experiment, late endosomes and lysosomes were fluorescently labeled in MNP-loaded HeLa cells under magnetic attraction, to verify the identity of magnetic endosomes (Appendix A’). Regarding the endo-compartment in which MNPs remain, as expected, internalized nanoparticles are delivered to the lysosomes during the time of magnetic attraction (Appendix A) and are not found in late endosomes (Appendix A).

These results are encouraging, demonstrating that the particles are internalized and delivered to the lysosomes by following normal kinetics without altering their short-term viability. In addition, we saw that the fluorescence of the particles does not seem to be affected after 10 days in growing cells, demonstrating their remarkable stability in a hostile acidic environment such as the lysosome (Appendix A).

### 2.3. Nanoparticle-Mediated Forces Direct Migration of Adherent Cells and Can Trap Them

Next, we assessed whether we could remotely control cellular functions such as cell migration using our magnetic tool. The spatial manipulation of cells was previously achieved in non-adherent conditions [39], but the actuation of cell migration in adherent conditions is much more challenging. We performed our first migration assay with HeLa cells, a robust cell that can endure strong magnetic constraints. Cells loaded with particles and plated on the MMA were imaged in transmission and in fluorescence every 5 min for 36 h under magnetic activation. Our results clearly show that pillars can exert substantial forces on the endosomes, attracting them towards the cell membranes and thus triggering migration-like behavior, such as polarization and displacement in the direction of the micro-pillar (Figure 2a). In addition to this directed “migration”, we could also observe prolonged trapping of living cells for several hours at the magnetic pole and transient trapping followed by an escape. Regarding the trapping, the different behavior of cells might be related to the varying MNP loading of the cells, a parameter that unfortunately cannot be controlled. Among all experiments, we observed robust attraction of the cells towards the magnetic poles of the micro-pillar, whereas no significant attraction was observed in the region orthogonal to the direction of the magnetization, where the magnetic field gradient was minimal (thus this direction can be considered as the no-force control). The mean fluorescent intensity (from the MNPs contained within the cells, which could be MNP signals and thus could be regarded as a cell marker) was then averaged over the final 24 h of the magnetic attraction period at the magnetic poles, at the non-magnetic poles of each micro-pillar and on a defined area surrounding the whole pillar with its attraction zone (*n* = 19 poles; Figure 2b,c). It clearly shows that, during the magnetic attraction, fluorescent endosomes filled with particles accumulate locally at the magnetic poles with an enrichment of four at the magnetic pole (*n* = 19; Figure 2d), confirming the successful control of cell attraction and retention by the magnetic poles of the micro-pillars. (Appendix A).

The same 24-h magnetic experiment and analysis were performed with neuron-like cells, undifferentiated SHSy-5Y. Additionally, for these cells, imaging was also performed several hours after switching OFF the magnetic field, and cellular magnetic relaxation was estimated by measuring fluorescence intensity. As was the case for HeLa cells, SHSy-5Y cells were responsive to the mechanical tension generated by magnetic endosome accumulation at the cell membrane. However, in contrast to HeLa cells, SHSy-5Y cells were attracted toward the magnetic pole in a collective manner and a higher accumulation of cells in time was measured (Figure 3). A higher proportion of trapped SHSy-5Y cells were accumulated over time, indicating that cellular escape was less probable for SHSy-5y than for HeLa cells and cellular capture was thus more efficient. As a matter of fact, the mean cellular enrichment at the magnetic pole was around six for SHSy-5y cells, whereas it was less than four for HeLa cells. If we compare the enrichment ratio between magnetic and non-magnetic poles, SHSy-5y cells responded twice as well as HeLa cells. This difference might be explained by the fact that SHsy-5y are smaller cells that are only loosely attached to the substrate, and tend to form colonies that will thus be more sensitive to the magnetic force. Moreover, this different behavior might be related to the intrinsic ability of cells to adopt directionally persistent migration, an ability that can be different from one cell type to another, and which is related to the cellular migration speed. Indeed, characteristics of motility, such as speed and persistence, are diverse and dependent on the cell type, origin and external cue [40]. HeLa cells might be possibly not as persistent and as fast as SHSy-5Y cells, so are more likely to undertake random migration, enabling them to escape the magnetic trapping. Finally, the lower trapping effect observed in HeLa cells could also be explained by a more heterogeneous MNP loading, thus allowing weakly loaded cell populations to escape the magnetic attraction. Interestingly, after switching OFF the magnetic field, a majority of trapped SHSy-5Y cells moved away from the micro-pillar and restarted random migration. This reversible cellular capture demonstrates that cells can survive after 24 h of magnetic constraint. As for HeLa cells, the likely capture of cells at the magnetic pole caused a slight cellular depletion at the non-magnetic side of micro-pillar (Appendix A).

Altogether, HeLa cells and SHSY-5Y cells can be remotely attracted by magnetic forces and can be captured reversibly during a long period of time at the magnetic poles of the micro-pillars.

### 2.4. Toward the Parallelized Magnetic Manipulation of Neurite Outgrowth

After these promising results on migration, we investigated how the mechano-actuation approach could control the growth of neuronal processes. Given the wide range of forces possibly generated by our system, we assumed that lower force-mediated neurite elongation would be possible to observe. Indeed, as the amplitude of the force depends both on the MNP loading and on the positions of the cells relative to the micro-pillars, statistically we should be able to observe cells that are in the appropriate force range among the cell population plated on the MMA. Fresh cortical neurons were loaded with MNPs and plated on an MMA pre-coated with laminin to stimulate axonal growth. Images of 10 different positions were taken in transmission and fluorescence every 5 min for 24 h (Figure 4a,b; Appendix A). One single position showed an interesting cellular response, confirming the narrow range of forces that need to be reached to target specific processes. At this distance, we observed two events of directed neurite outgrowth followed by directed cell attraction (Figure 4a). Interestingly, the attraction of these two cells toward the magnetic pillar was preceded by protrusion formation and neurite elongation. These two cells not only migrated in the direction of the micro-pillar, but they also grew neurites in alignment with their movement. Even more interestingly, for one of the two cells, a larger size neurite was formed on the front of the cell, possibly the future axon [41], while a smaller size process was visible on the back. Both directed migration and neurite outgrowth processes may have occurred simultaneously, giving evidence that polarity can be controlled through MNP-induced mechanical forces. However, the forces we apply would have to be modulated properly and the experiments reproduced in order to demonstrate statistically the directed axonal outgrowth without cell body displacement.

Another similar experiment was performed for a longer period of time (72 h). Images of 22 pillar positions were taken in transmission and fluorescence every 10 min for 72 h (Appendix A). Only one single clear event of directed neurite outgrowth, without cell migration, could be observed among all the positions recorded (Figure 4b). Even though this neuron already exhibited two spontaneous neurites in the orthogonal direction to the magnetic field, it was observed that a third neurite initiated and grew towards the micro-pillar. The first step (elongation) was characterized by its fast speed, estimated around 0.12 mm/h^−1^, and by the thinness and poor adhesive properties of the neo-formed tube. After elongation ended, the process was followed by the thickening of the membrane tube, coupled with an intense stretching of the cell body. These observations are consistent with a study proposing that axons thinned during high rates of elongation and thickened when the growth cones were stationary [42]. Moreover, the visible flow of MNP-filled endosomes from the cell body to the fiber tip attested to the fiber membrane’s integrity. Even though the speed of spontaneous axon growth observed in vitro is estimated at 0.2–1 mm/day in the case of neurons of the central nervous system [43,44,45,46], the maximum stretching rate that a neuron of the peripheral nervous system could tolerate has been measured at 8 mm/day without impeding its integrity and its ability to propagate an electrical signal [45,47]. Thereby, with a rate of 3 mm/day, the magnetic stretching observed is physiologically acceptable to produce a functional axon. The prolonged survival of the neuron exhibiting an artificial neurite (18 h) attests to the biocompatibility of this MNP-mediated method. Altogether, our results provide a proof-of-concept that mechanical forces produced by magnetic endosomes can induce the directional initiation and growth of a neurite, and possibly of an axon, at an accelerated but nonetheless physiological rate.

### 2.5. Magnetic Endosomes Can Induce and Direct Neurite-Like Tubes from Pc12 Cells but Cannot Guide Preexisting Neurites

As differentiated PC12 cells displayed natural accumulation of MNP-filled endosomes in the tips of their branches (Appendix A), this cell line appeared us to be a very good model to demonstrate force-induced neurite elongation. Indeed, the distribution of magnetic endosomes at the neurite tip in the absence of magnetic forces makes possible the pulling and guidance of the extremity of a process, akin to the performance of a growth cone. Unfortunately, multiple magnetic experiments with PC12 cells did not show any directed neurite elongation, nor any cellular attraction from cells exhibiting MNP accumulation in their neurite tip. On the contrary, the differentiated PC12 cells rather retracted their processes, with or without accumulation of MNP at their tip, under the effect of the magnetic field. (Figure 4c, View 2; Appendix A). However, we could observe on several cells lacking branches the elongation of one single neurite-shaped membrane tube in the direction of the micro-pillars (Figure 4c). These membrane tubes cannot really be identified as formal neurites due to their small cross-sectional diameter, their wave motion and their poor adhesion (Views 1 and 2; Appendix A). Similar properties have been observed in neurons at the early stage of the growth of magnetically induced neurites, characterized by a rapid rate of elongation. These results suggest that intracellular magnetic endosome distribution, either dispatched between the cell body and branches or concentrated only in the cell body, may be crucial to achieving the threshold of forces needed for initiating membrane protrusive processes. The retraction effect due to the magnetic endosome manipulation has already been reported in a previous study, in which it was shown that neurite outgrowth can be put on hold by applying forces in the low piconewton range produced by magnetic endosomes [48]. Thus, opposite effects, such as elongation or retraction, might be triggered according to the strength of forces applied locally to the plasma membrane, which directly depend on the local density of magnetic endosomes. However, we cannot exclude that the retraction is a cellular stress response due to the magnetic field exposition or experimental conditions. Manipulating endosomes filled with MNPs may therefore be a good method of remote-control initiation and growth of a neurite in a given direction at an early stage of differentiation.

## 3. Discussion

In this study, we have engineered a biocompatible micro-magnetic array (MMA) capable of imaging multiple cells at the same time over several days. Using this MMA, we show evidence that a mechanical approach using biocompatible SPION-mediated forces is a viable physiological method to activate directional migration or neurite elongation, while preserving vital cell functions. We confirmed that forces in the pN range are involved in the transport of organelles, protrusion formation, neurite elongation and cell migration. Our findings show that MNP-mediated forces have a strong impact on directional cell motility, and can be used to trap adherent cells reversibly on magnetic pillars. We also found that the level of control was dependent on the cell type, most likely in relation with MNP uptake distribution among the cell population, together with the cellular adhesion, size, shape and motility. Indeed, neuronal SHSy-5Y cells, growing in a grouped manner, with a small size and fast rate displacement, responded much better to the applied magnetic forces than larger, slower HeLa cells, which avoid contact with surrounding cells.

Considering cell migration, use of the MMA was essential to allow the collection of larger data sets of statistical relevance, and thus to assess robust effects on highly variable cellular behaviors. For application to neurons, we have used this same magnetic array as a platform to screen magnetic force ranges based on the distance of cells to the pillars, thus offering the opportunity to observe rare events that require precise magnetic conditions. We could thus observe just a few instances of neurite outgrowth with neurons, despite the fact that experiments were highly parallelized, involving several hundred cells. From the experimental design, we were then able to infer the correct range of forces, which we estimate to be around 5 pN, required to actuate neurite outgrowth without promoting the attraction of cell bodies. With a comparative approach, Kunze et al. found that MNP-mediated magnetic forces were able to relocate intracellular tau protein distributions (an axon marker) and thus had the ability to redirect neuron polarity. In regard to this study, it is likely that the induced processes we observed in our magnetic experiments are axons at an early stage. Importantly, we also found, as a preliminary observation, that weak pulling forces acting at the tip of pre-existing neurites did not promote elongation, but instead induced retraction. We propose that a threshold of forces needed to promote protrusive processes likely occurs in cells lacking pre-exiting neurites, with a more efficient magnetic endosome recruitment onto the cell body plasma membrane, compared to star-shaped cellular structure where the magnetic endosomes are dispatched between the soma and branches.

Raffa et al. revealed the importance of generating low piconewton-range forces along the whole axon to reproduce natural stretch growth. To achieve this stretching, MNPs in this prior study were shown mainly to escape from endo-compartments and to exert extremely weak forces (pN-range) throughout the cell. In our case, based on the measured fluorescence distribution, Fe_2_O_3_-PAA_2k_ nanoparticles were mostly located in endo-compartments after one day of MNP incubation for SHSy-5Y cells (Appendix A) and remained in lysosomes in the first 24 h of magnetic attraction for HeLa cells (Appendix A’). Similarly, TEM images suggest that the majority of MNPs remain within endo-compartments (Appendix A). As the cell type and MNP characteristics (size, shape, surface chemistry) are decisive in the endocytosis pathway taken, it is possible that MNPs show different intracellular MNP distributions from one study to another. Based on the forces we have estimated (see Materials and Methods), tens to one hundred pN forces were applied in our migration assays. Regarding the smaller size of neurons and the fast decrease of MNP loading over magnetic attraction for PC12 cells, we assumed that lower forces were exerted in neurite outgrowth experiments (below 10 pN). The strategy that we employed is different from that of Raffa et al., since the mechanism of action of the force is not the same. Though Raffa et al. promoted directed axonal stretching, we instead accumulated local tension onto the membrane and at the neo-formed neurite tip (based on the accumulation of MNPs seen in the fluorescence measurements) and observed directed initiation and growth in the same manner as the natural growth cone’s exertion of pulling forces. To determine if this pulling strategy is adaptable to nerve regeneration, results should be reproduced more statistically. First and second magnetic devices should be adapted to apply a force of 1–5 pN over long distances. For regenerative medicine, the design requirement for such a device is a uniform field gradient of 10^3^ T/m over a centimeter length scale, together with a homogeneous magnetic field above 150 mT in order to saturate the nanoparticles. Although obtaining a homogenous field is readily achievable, generating high magnetic field gradients over long distances is challenging. Using a permanent magnet smaller than 1 mm (to achieve gradients as high as 10^3^ T/m), the effective span of attraction is about ~100 µm, still quite small for therapeutic applications. Future arrangement of macroscopic magnets such as Halbach arrays may increase the span of required forces up to centimeters. It is also possible that smaller gradients, around 10–100 T/m, are sufficient to actuate cellular processes, but on a longer time scales than those we considered in this work. However, at the present time there is no practical solution that enables magnetic actuation of nanoparticles over the length scale of tens of centimeters. Thus, our application for regenerative medicine might be more adapted for spinal cord injury, for example, where the extent of the magnetic attraction required is smaller than for brain disorders.

## 4. Materials and Methods

### 4.1. Synthesis of Fe_2_O_3_-PAA_2k_-Rhodamine Magnetic Nanoparticles (MNPs)

Maghemite γ-Fe_2_O_3_ nanoparticles were synthesized by co-precipitation of iron II and III chloride salts in basic conditions. The obtained nanoparticles were then size-sorted by gradually increasing the ionic strength of the solution with nitric acid and thus flocculating the largest nanoparticles. At the end of the magnetic core synthesis, a dispersion of 8.4 ± 1.9 nm γ-Fe_2_O_3_ nanoparticles at a concentration of 9.09 wt% in nitric acid at pH = 2 was obtained.

In order to render the nanoparticles fluorescent, the maghemite cores were functionalized with rhodamine B. A total of 0.3 g of amino-PEG-phosphonic acid hydrochloride (phosph-PEG-NH_2_, Mw = 2100 g/mol, Specific polymers, Castries, France) was dissolved in 59.7 g of deionized (DI) water. The pH of the solution was brought to 8.5 with 30% ammonia. Then, 1.53 mL of a 10-mg/mL solution of rhodamine B isothiocyanate was added to the phosph-PEG-NH_2_ solution. The solution was agitated for two days and the pH of this phosph-PEG-Rho solution was finally brought to 2 by the addition of nitric acid. Then, 57.5 g of this solution was added to 81.9 g of the γ-Fe_2_O_3_ nanoparticle dispersion at 1 wt%. After 10 min of agitation, the nanoparticles were slowly added to 174 g of a solution of poly(acrylic acid) (PAA_2k_, Mw = 2000 g/mol) at 1 wt% at pH = 2. The MNPs were then magnetically removed from the solution and redispersed in a 20% ammonia solution. They were then dialyzed against DI water 5 times and against water at pH = 7.5 once.

### 4.2. Characterization of MNPs

The MNPs were characterized (Appendix A) by transmission electron microscopy on a JEOL 1011 instrument. The mean physical diameter of the MNPs, 8.4 ± 1.9 nm, was determined by measuring 330 single particles using the software ImageJ. The MNPs were also characterized by dynamic light scattering on a Malvern Zetasizer Nano ZS instrument. The average hydrodynamic diameter of the Fe_2_O_3_-PAA_2k_-rhodamine MNPs in water at pH 7.5 was 27.4 nm, with a polydispersity index of 0.155, whereas in cell culture medium supplemented with 10% serum, the mean hydrodynamic diameter of the Fe_2_O_3_-PAA_2k_-rhodamine MNPs was 30.1 nm, with a polydispersity index of 0.223. Finally, the magnetic properties of the MNPs were analyzed by superconducting quantum interference device (SQUID) magnetometry on a Quantum Design MPMS-XL instrument at the Physical Measurements at Low Temperature Platform (MPBT) of Sorbonne Université. The MNPs are superparamagnetic, with a saturation magnetization of 55.7 emu/g, and the magnetic diameter of the particles was measured with a Langevin fit to be 7.96 nm (*σ* = 0.26).

### 4.3. Cell Culture

The human cervical cancer HeLa and the human neuroblastoma SH-Sy5Y cell lines obtained respectively from ATCC and from Ruhr-Universität Bochum, Germany were cultured in Dulbecco’s modified Eagles’s medium/F12 mixture containing 2 mM L-glutamine (Life Technologies, Villebon-sur-Yvette, FR), 10% fetal bovine serum (FBS, Eurobio, Les Ulis, FR) and 1X penicillin/streptomycin (Life Technologies, Carlsbad, CA, USA). Rat pheochromocytoma PC12 cells (Ruhr-Universität Bochum, Bochum, Germany) were cultured in Dulbecco’s modified Eagles’s medium high glucose (Sigma-Aldrich, Saint-Quentin-Fallavier, FR) containing 10% horse serum (Sigma-Aldrich, Saint-Quentin-Fallavier, FR), 5% fetal bovine serum (Eurobio, Les Ulis, FR) and 1X penicillin/streptomycin (Life Technologies).

Cells were cultured in 75 cm^2^ flask without coating for HeLa and SH-Sy5Y cells or with 10 µg/mL poly-D-lysine coating (PDL, Sigma-Aldrich, Saint-Quentin-Fallavier, FR) for PC12 cells and maintained at 37 °C in a saturated humidity atmosphere of 95% air and 5% CO_2_. HeLa, SH-Sy5Y and PC12 cells were used at passages 15–20.

For cell differentiation, PC12 cells were incubated in serum-reduced medium (1% horse-serum, 1% FBS) supplemented with 100 ng.ml^−1^ ß-NGF (Sigma-aldrich, Saint-Quentin-Fallavier, FR) during five days on the PDL-coated surface.

Cortical neurons were prepared from cortical hemispheres dissected from C57BL6N mouse embryo brains (E15-16). After rinsing with Gey’s Balanced Saline Solution (Thermo Fisher, Waltham, MA), tissues were dissociated in Dulbecco’s modified Eagle’s medium (DMEM; Thermo Fisher, Waltham, MA), Villebon-sur-Yvette, FR) containing 25 mg/mL papain (Sigma-Aldrich, Saint-Quentin-Fallavier, FR) for 10 min at 37 °C and the enzymatic reaction was stopped by addition of 10% FBS (Thermo Fisher). The digested tissues were then mechanically dissociated with a pipette in DMEM containing DNase I (Sigma-Aldrich). Cells were washed once by centrifugation (80× *g* for 6 min) and resuspended in complete Minimum Essential Medium (MEM; Thermo Fisher) supplemented with 5% FBS (Thermo Fisher), 1% N-2 (Thermo Fisher), 2% B-27 (Thermo Fisher), 2 mM Glutamax (Thermo Fisher), 1 mM sodium pyruvate and 20 µg/mL gentamicin (Thermo Fisher).

Cortical neurons were plated on glass surface coated with 10 µg/mL PDL in complete MEM then cultured and imaged in Neurobasal (Thermo Fisher) with 1% FBS, 2% B27, 2 mM glutamax and 20 µg/mL gentamicin on the surface, coated with 10 µg/mL PDL and 4 µg/mL laminin (Sigma-Aldrich).

### 4.4. Cell Transfection for Endosome Labeling

Lysosomes and late endosomes were labeled separately in HeLa cells by transfection with pEGFP-Rab9 or pLL7-IRFP-Rab6a plasmids, respectively. For this process, HeLa cells were plated one day before in a 6-well plate at 50% confluency and transfected with 1 µg plasmid and 3 µL XtremeGeneHP transfection reagent (Sigma-Aldrich, Saint-Quentin-Fallavier, FR) for 24 h. Cells were then loaded with MNPs, plated on the micro-magnetic array and imaged under a magnetic field, as described below.

### 4.5. MNP Loading by Passive Endocytosis

HeLa, SHSy-5Y and PC12 cells were seeded one day before at high density in a 6-well plate, more precisely with 200,000 cells, 400,000 cells or 500,000 cells per well, respectively.

Fresh dissociated mouse cortical neurons were seeded 3 h before MNP loading at high density in 6-well plate, on 25 mm round glass coverslips (#1, Thermo Fisher) pre-coated with 10 µg/mL PDL (Sigma) in complete MEM with 2 million cells per coverslip.

Following the internalization procedure described in Safi et al., 2011, tested with success for HeLa cells, Fe_2_O_3_-PAA_2k_-rhodamine MNPs were added at 10-mM final concentration in complete medium for 24 h. For the more sensitive SHSy-5Y, PC12 cells and cortical neurons, MNPs were incubated at the lower final concentration of 2 mM for 18–24 h.

### 4.6. Micro-Magnetic Array Fabrication and Characterization 

Micro-magnetic arrays, or MMAs, consist of a matrix of soft ferromagnetic elements, or micropillars, that can be magnetized by an external field. The soft ferromagnetic elements are made out of permalloy, which is a nickel-iron alloy known for its interesting magnetic properties. This material gets fully magnetized very fast and has a very low hysteresis curve, meaning that in the absence of an external field the magnetization of the material goes back to zero, allowing for an on/off magnetic switch. The device was made using conventional microfabrication techniques as previously described ([36], Appendix A). The biocompatible layer is processed on top of the magnetic array, such as a layer of PDMS that is spin-coated with a thickness approximately equal to the height of the pillars.

### 4.7. Magnetic Control Experiments on Micro-Magnetic Array (MMA)

PDMS-coated micro-magnetic arrays were cleaned in 70% ethanol, dried under a sterile hood, incubated for 30 s in plasma cleaner, then coated with cell-specific adhesive surface—for at least 1 h with 10 µg/mL recombinant human fibronectin (Sigma) for Hela and SHSy-5Y cells, for 15 min with 10 µg/mL poly-D-lysine for PC12 cells or overnight with 10 µg/mL PDL followed by at least 1 h incubation with 4 µg/mL laminin (Sigma) for cortical neurons.

Cells loaded with MNPs were washed carefully 3 times with 37 °C pre-warmed PBS to remove free MNPs in the medium and were detached under action of trypsine for 5 min at 37 °C. Centrifuged cells were resuspended in complete medium, counted and plated at low density on the pre-coated micro-magnetic array. More precisely, 50,000 HeLa cells, 100,000 SHSy-5Y or 500,000 cortical neurons were seeded on the array in complete medium for at least 3 h. With the exception of PC12 cells that were kept for 5 days in differentiation medium before imaging under the magnetic field, HeLa, SHSy-5Y and cortical neurons were imaged after replacing medium, either with complete DMEM/F12 medium for HeLa and SHSy-5Y or with complete neurobasal medium for cortical neurons. After complete adhesion of cells on the substrate, the MMA containing cells loaded with MNPs was ready to be installed in a specifically designed chamber, suitable for culture, imaging and magnetic activation. Microscopy and digital image acquisition were carried out with a Nikon Eclipse TE2000-s or Olympus IX71 inverted microscopes, driven with Metamorph software. Multiple position time-lapse acquisitions in both transmission and fluorescence were launched under magnetic activation over 24–72 h. A homogeneous 100 mT external field on the whole surface of the MMA was applied by the simultaneous installation of two external square magnets on both sides of the array (N pole facing S pole) that allowed the full magnetization of all pillars and magnetic nanoparticles (such that a gradient of 10^3^ T/m was generated 10 µm away from two sides of the pillars, Figure 1a).

### 4.8. Image Analysis and Statistic Evaluation

To analyze the magnetic control of migration experiments performed on HeLa and SHSy-5y cells, the cellular distribution at both magnetic and non-magnetic poles of each micro-pillar was evaluated by measuring the fluorescent signal of MNPs along the four edges of every pillar. A window of 15 µm around each edge was used to estimate the mean fluorescence. We assumed that the fluorescence intensity corresponds to the amount of MNP uptake by the cells, such that the total signal was a quantitative proxy of cell density. To quantify the mean fluorescence intensity averaged over time, stacks of time-lapse fluorescent images were background subtracted and then Z-projected in time (using “average intensity” in ImageJ). The ratio of the time-projected mean fluorescence intensity (MFI) between the 15 µm-proxi-regions of micro-pillar (V and H) and the total area (B) was plotted to estimate the magnetic enrichment of cells at the magnetic poles. The two-tailed *p*-value was calculated in Graph Prism using unpaired tests with Welch’s correction.

### 4.9. Force Calculation

In order to estimate the total force acting on the cell, we computed the force produced by a single nanoparticle (1), by a magnetic endosome (2) and by all the endosomes or lysosomes, assuming that all endosomes eventually collapse in lysosomes (3). The force acting on a single nanoparticle, assuming that the nanoparticle is saturated (a 100-mT field is sufficient given the measured magnetization curve; see Appendix A), is given by:(1)$$F=\rho VM_{s}10^{-3}\nabla B≅0.35\;\mathrm{fN}\;(10\;µm\;\mathrm{distance})$$
where F is the force produced in Newtons, *ρ* is the density of the maghemite core (5100 kg/m^3^), V is the volume of the 8.4 nm magnetic core (3.10339E ^−25^ m^3^), Ms is the magnetization of the nanoparticle at saturation (50 000 emu.kg ^−1^), 10 ^−3^ is the conversion factor from emu to A.m^2^ and ∇B is the magnetic gradient (10^4^ N/(A.m^2^)). The force per endosome is then
(2)$$F_{endo}=N_{part}\times F≅0.31\;\pm 0.06\;\mathrm{pN}\;(10\;µm\;\mathrm{distance})$$
where $N_{part}$ is the number of nanoparticles in one endosome. Assuming that nanoparticles occupy half the volume of the endosome (see TEM images, Appendix A) we obtained *N_part_* = 991 ± 378 for endosomes of sizes between 400 nm and 600 nm. This number is in accordance with direct counting of the nanoparticle number in 100-nm slices of TEM images.

Eventually, it becomes difficult to know how many magnetic endosomes are contained in every cell, especially since the efficiency of internalization varies from one cell type to another. According to the TEM images (Appendix A), SHSy-5Y cells contain between 72 ± 30 endosomes per cell, thus the cumulative force of all the endosomes acting on the cell membrane or on the cell cytoskeleton would be:(3)$$F_{cell}=N_{endo}\times F_{endo}≅22\pm 16\;\mathrm{pN}\;(10-µm\;\mathrm{distance})$$

This order of magnitude is consistent with the work of Kunze et al. and Raffa et al.

## 5. Conclusions

In this work, we have shown that several cell behaviors can be remote controlled using passive internalization of magnetic nanoparticles and switchable micro-magnetic arrays. Adherent cells can be attracted toward the micro-magnets and reversibly trapped on their magnetic poles. With primary neurons, we gave a proof of concept that outgrowth can be initiated and guided by magnetic forces. We found evidences that cellular actuation is done through forces applied by endosomes filled with magnetic nanoparticles. The exact mechanism by which it happens, whether forces are applied on the plasma membrane, on the cytoskeleton, or if forces are transduced into signaling events remains to be elucidated. For future therapeutic applications, we highlighted that the range of forces needs to be precisely optimized and that the effective span of actuation is limited to relatively small distances. 

## Figures and Tables

**Figure 1 ijms-21-06560-f001:**
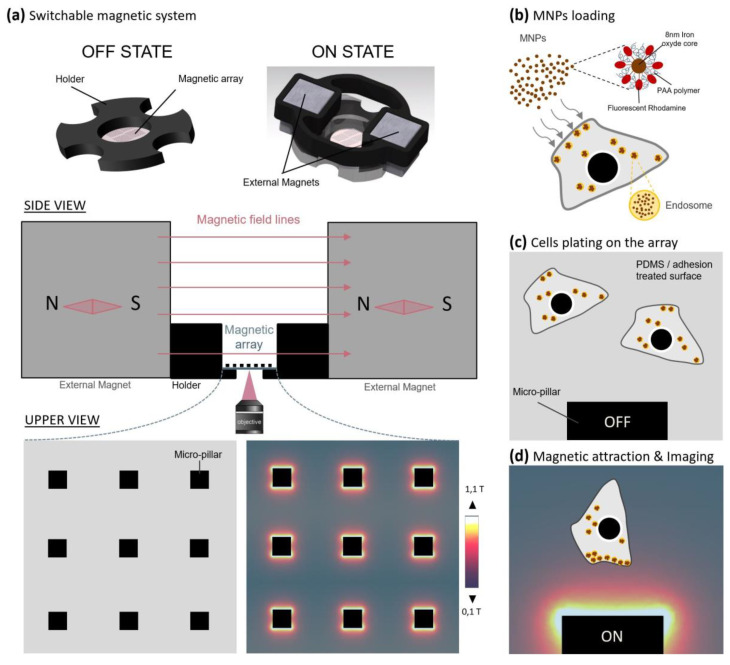
Parallelized magnetic manipulation on a micro-magnetic array. The micro-magnetic array consists of a matrix of 100-µm-wide soft ferromagnetic pillars. The micro-pillars can be magnetized using a homogeneous external magnetic field (ON State) or demagnetized when the field is removed (OFF State). Magnetized micro-pillars generate a high magnetic gradient up to 10^4^ T/m (**a**). The micro-magnetic array was made biocompatible by coating the surface with thin PDMS layer and an adhesion molecule coating. To perform the magnetic manipulation experiment, cells were first loaded with magnetic nanoparticles (MNPs) during 18–24 h in 6-well plate using the passive endocytosis process (**b**). Loaded cells were then plated on the magnetic array until their complete adhesion without magnetic activation (OFF state) (**c**). Finally, external magnets were added in both sides of the array to activate the micro-magnetic pillars (ON state) and both transmission and fluorescence imaging were done over 18 h–72 h in a 37 °C 5% CO_2_ chamber (**d**).

**Figure 2 ijms-21-06560-f002:**
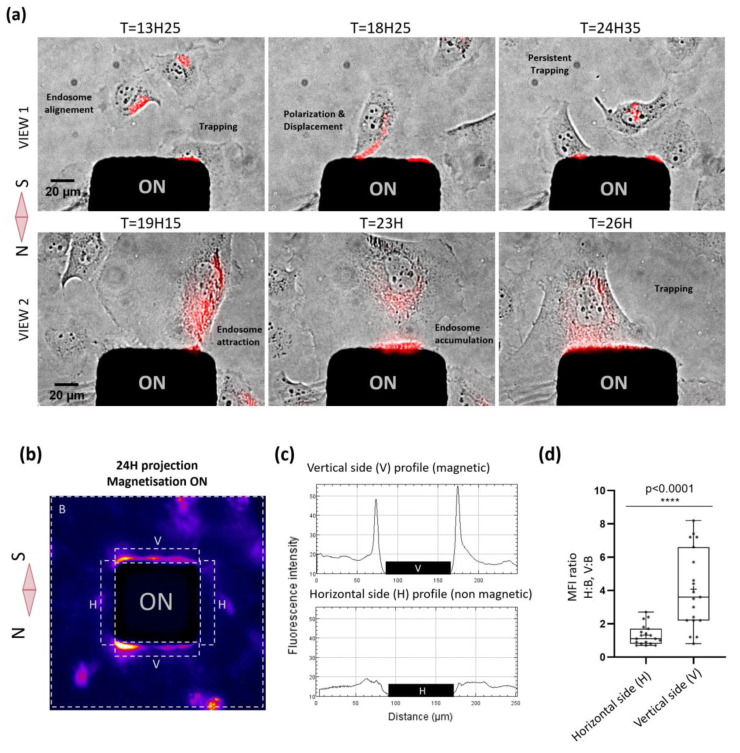
Parallelized magnetic manipulation of HeLa cells. (**a**) Images of 2 representative views at different times of magnetic attraction, showing cellular responses, such as polarization, displacement or trapping toward the magnetic pole, consequently to the attraction and accumulation of the magnetic endosomes, depicted in red. (**b**) Averaged images (*n* = 19) of mean fluorescence intensity time projection comprising the different areas measured to estimate the magnetic cellular trapping: V (vertical magnetic pole), H (horizontal non-magnetic pole) and B (background total area). (**c**) Plots representing the Fluorescent intensity profile at the magnetic pole (vertical side) and at the non-magnetic pole (horizontal side) of the micro-pillar. (**d**) Histogram comparing the cell enrichment in the region close to the magnetic pole (V) and close to the non-magnetic pole (H, control). MFI = mean fluorescence intensity.

**Figure 3 ijms-21-06560-f003:**
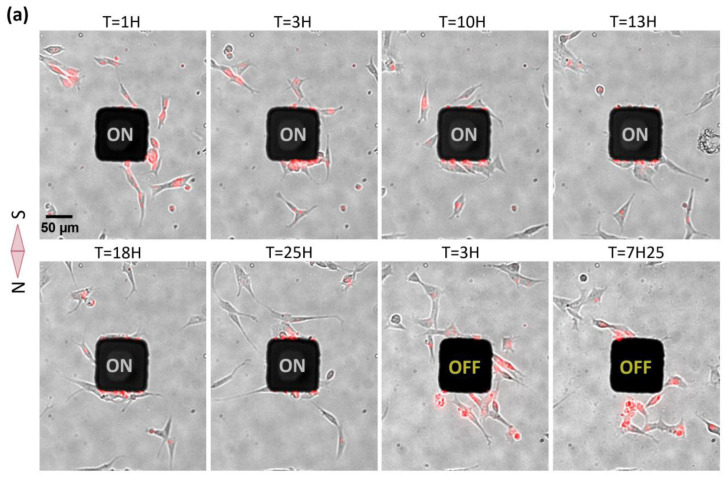
Parallelized magnetic manipulation SHSy-5y cells. (**a**) Images of one representative field of view at different times during magnetic attraction (24 h), showing collective cellular trapping at the magnetic pole, consequently to the attraction of the magnetic endosomes, depicted in red. After removing the magnetic field (OFF State), captured cells are released and move away from the pillar (last two images). (**b**,**d**) Average images (*n* = 25) of mean fluorescence intensity time projection comprising the different areas measured to estimate the magnetic cellular trapping and relaxation: H (horizontal axis, control), V (vertical axis with and without an external field) and B (background area). (**c**,**e**) Curves representing the fluorescent intensity along V and along H. (**f**) Histogram comparing the cell enrichment between the region close to the magnetic pole (V) and the region orthogonal to the magnetic poles, where no magnetic field gradient is produced ((H region, control)). MFI = mean fluorescence intensity.

**Figure 4 ijms-21-06560-f004:**
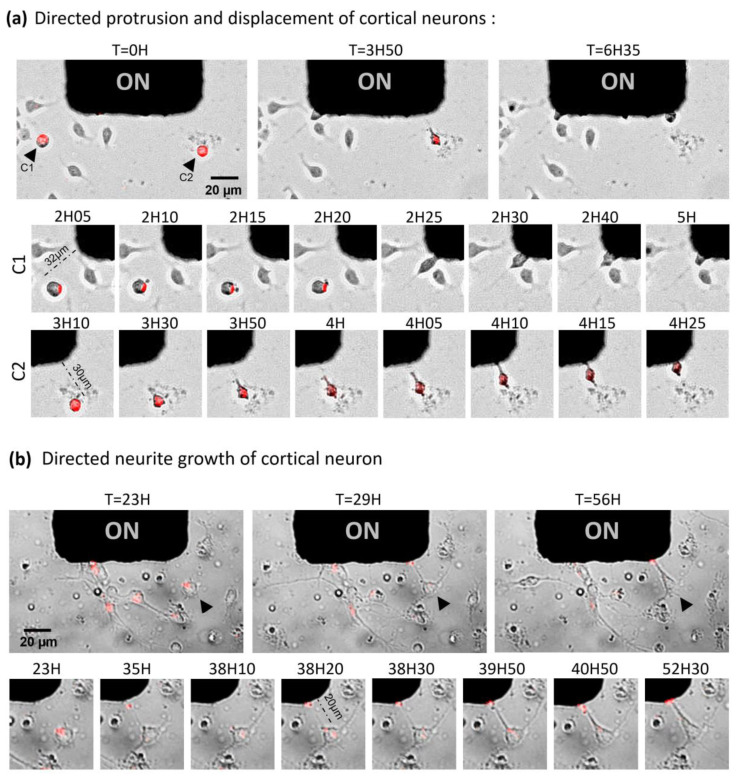
Towards the parallelized magnetic manipulation of neurite outgrowth. (**a**) Images of cortical neurons loaded with magnetic nanoparticles (in red) during 24 h of magnetic attraction. At time 0 h, the black arrows show two neurons containing magnetic particles. At 2 h 10 min, a protrusion formed towards the pillar and co-localized with the particles’ enrichment region (red dot). At 2 h 25 min, this same cell (left) polarized and migrated towards the magnetic pillar. The speed of the displacement was estimated to be 190 µm/h. At 3 h 30 min, the right cell polarized, exhibiting the characteristic triangular shape preceding cell migration or axonal growth. From 3 h 50 min to 4 h 10 mins, a neurite form lengthened and aligned with the movement of the cell towards the pillar. At 4 h 25 min, the cell completed its movement against the pillar. The movement speed was estimated to be 30 μm/h. (**b**) Images of cortical neurons loaded with magnetic nanoparticles (in red) placed under a magnetic field for 72 h. Among 22 pillar positions, one single neuron, shown by the black arrow, displayed a directed neurite outgrowth towards the micro-pillar without cell migration. At 35 h, a tube with MNPs at its tip formed very quickly and briefly toward the magnetic pole, before vanishing. At 38 h 10 min, we observed a neurite growth towards the micro-pillar at a speed of 120 µm/h with the presence of MNP accumulation (red spot) at the tip, touching the magnetic pole. The elongation was followed by the thickening of the neurite, which remained in place for 18 h with good viability of the neuron and despite the cellular body deformation due to the strong pulling. (**c**) Images of PC12 cells, loaded with magnetic nanoparticles (in red), treated for 5 days with nerve growth factor (NGF), during 24 h of magnetic attraction. Several cells displayed neurite-like tubes growing directionally towards the micro-pillar and loosely attached to the surface. Two of them are shown in this figure in views 1 and 2 (see left black arrow in view 2). Multiple cells with MNP accumulation in the tip of spontaneous minor processes showed retraction of the fiber, rather than elongation. One event is shown in View 2 (right black arrow).

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
