# Peer review of "Parallelized Manipulation of Adherent Living Cells by Magnetic Nanoparticles-Mediated Forces"

_ijms, 2020, doi:10.3390/ijms21186560_

Round 1
Reviewer 1 Report
Smart magnetic nanomaterials belong to intensively studied materials especially for bioapplications. The possibility to control them at distance by external magnetic fields made promising for therapeutic applications. In this paper the authors present biocompatible micro-magnetic array capable of imaging multiple cells at the same time over several days. The obtained results bring an important challenge in fundamental research as well as in regenerative medicine.
The Supplementary videos are very impressive.
Recommendation:
Description of some Figures is very difficult to read, especially Figs S1 and S2
Author Response
We thank reviewer 1 for its positive review. We rewrote the legends of Figs S1 and S2 for clarity (we added a title as well for these figures).
Reviewer 2 Report
In this paper, the authors combined micro-magnetic array with MNPs to actuate cellular migration and neuritis outgrowth. The fluorescent MNPs are used for simultaneous magnetic force manipulation and fluorescence imaging four kinds of cells in vivo. The stability and biocompatibility of MNPs living cells are well-demonstrated and results are convincing. The cellular behaviors are well explained by the MNP induced forces and fluorescent images confirming the theories. Thus, I suggest acceptance of this paper after the authors making a minor change to it:
Line 474, some numbers are missing: “the effective span of attraction is about ~,”
Author Response
We thank reviewer 2 for its positive review. We have corrected Line 474.
Reviewer 3 Report
The authors Bongaerts and colleagues report the development of smart magnetic nanomaterials for manipulation of neurite cells using micro-magnetic pillars. In particular a parallelized magnetic tool was used to promote migration and neurite outgrowth by applying mechanical tension on PC12 cells and mouse primary cortical neurons. The results showed the stability of magnetic nanoparticles into the cells lysosomal compartment and their impact on cells migration in adherent condition. Using the parallerized magnetic manipulation, the authors showed the ability of this device in promoting neurite outgrowth. Finally, the results showed the ability of magnenic endosomes in directing neurite-shaped membrane tube in PC12 cells but were not able to achieve the elongation of preexisting neuritis.
The paper introduce an innovative device as a candidate for potential biomedical purposes. This is an interesting work and has a good relevant to the field.
The manuscript would benefit from the following:
- In the introduction section lines 113-114 the authors should include the reference: Investigating the Mechanobiology of Cancer Cell–ECM Interaction Through Collagen-Based 3D Scaffolds. Cell Mol Bioeng. 2017 Mar 6;10(3):223-234. doi: 10.1007/s12195-017-0483-x. eCollection 2017 Jun.
- Assessment of nanoparticles toxicity should be more investigated. For example, metabolic activity, mitochoandrial damage, ROS production, oxidative stress, apoptosis and actin filament integrity could represent some valuable information.
- Limitation of the study should be included.
Minor revisions are requested
Author Response
We thank reviewer 3 for its positive review.
- We added the reference in the introduction as rightly suggested by reviewer 3.
- We agree with reviewer 3 that an exhaustive study of MNP toxicity would benefit for future therapeutic applications. However, such a study is beyond the scope of our work which is already substantial in terms of biological and physical characterizations. In addition, Fig S5 present survival of MNP loaded cells for almost 3 weeks. This is not a proof of a total absence of toxicity, but it clearly shows that cells can survive significantly. We also observed cell division (see for example Fig S5’ a) ) after MNPs uptake and under magnetic attraction, which suggest that cells behave normally in these challenging conditions.
- We think that we have already presented in details the limitations of the study. In particular, lines 436-438 we admitted that we could capture only few events despite the large number of cells and magnetic pillars; lines 443-445 we show that forces can induce retraction of fibers in some instances; lines 471-481 we discuss in details the limitation of our strategy for therapeutic applications.